# The Effects of an Order-Assist Mobile Application on Pediatric Anesthesia Safety: An Observational Study

**DOI:** 10.3390/children10121860

**Published:** 2023-11-27

**Authors:** Jung-Woo Shim, Chang-Jae Kim, Ji-Yeon Kim, Ji-Yeon Choi, Hyungmook Lee

**Affiliations:** 1Department of Anesthesiology and Pain Medicine, Seoul St. Mary’s Hospital, College of Medicine, The Catholic University of Korea, Seoul 02706, Republic of Korea; serendip3@catholic.ac.kr (J.-W.S.); 21902021@cmcnu.or.kr (J.-Y.K.); choi940207@cmcnu.or.kr (J.-Y.C.); 2Department of Anesthesiology and Pain Medicine, Eunpyeong St. Mary’s Hospital, College of Medicine, The Catholic University of Korea, Seoul 03341, Republic of Korea; ksw070591@catholic.ac.kr

**Keywords:** anesthesia, child, infant, medical errors, mobile applications, patient safety

## Abstract

Pediatric anesthesia requires the rapid creation, communication, and execution of anesthesia orders, and there is a risk of human error. The authors developed an order-assisted mobile application (app) to reduce human error during pediatric anesthesia preparation. The authors conducted an observational study that compared the effects of the application by comparing anesthesiologists’ errors, nurses’ errors, nurses leaving the operating room, and delays in surgery, between the Conventional group (*n* = 101) and the App group (*n* = 101). The app was associated with reduced human error by anesthesiologists and nurses, and it lowered the frequency and duration of nurses leaving the operating room during anesthesia. In addition, the authors surveyed anesthesia nurses regarding the effectiveness of the app. The nurses confirmed that the app was convenient and reduced human error. This study revealed that the order-assisted mobile app developed by a pediatric anesthesiologist could reduce human errors by anesthesiologists and nurses during pediatric anesthesia preparation.

## 1. Introduction

Human factors have been recognized as the primary cause of preventable patient harms [1]. They are critical elements of safe clinical practices. Human factors science is defined as the science of improving human performance and well-being by examining all the effectors of human performance [2]. In aviation, all aspects of the system, including knowledge, environment, and team dynamics, have been inspected and improved to minimize human error [3]. In the case of anesthesia, there are also possibilities of human errors at each step, from doctors creating orders and communicating them with the nurses to the actual application. However, preventing human error in anesthesia is entirely the responsibility of the anesthesiologist and the nurse in charge. This is much like only a few slices of cheese existing in the Swiss-Cheese model, and dangerous [4].

Depending on the treatment stage, medical errors can be categorized as diagnostic, therapeutic, preventive, or other. Diagnostic errors include those related to diagnosis and testing, while treatment errors are associated with surgeries, procedures, prescriptions, and medications. Preventive errors are related to preemptive treatment and follow-up. Other errors stand for communication, equipment, and systemic issues. The most common preventable errors fall into the “other” category, which includes technical issues. In pediatric anesthesia, patient characteristics, a busy operating room environment, and the burden on anesthesiologists and nurses can impact human errors [5].

According to an analysis of the reported medication errors during pediatric anesthesia in the Wake Up Safe database, incorrect dosing was the most common type of error [6]. Pediatric patients are particularly susceptible to anesthetic dosing errors because the dose needs to be adjusted based on the child’s weight and organ maturation. For the same reasons, children are more susceptible to incorrectly adjusted doses [7,8]. Thus, reducing human error is crucial in pediatric anesthesia.

Anesthesiologists and nurses are frequently exposed to distracting and high-pressure environments in the operating room (OR). In addition, they are often exhausted at work and human performance deteriorates rapidly in such environments [9,10]. Moreover, pediatric anesthesia is associated with more complications than adult anesthesia because children have less physiologic reserve and more vulnerable anatomies, such as higher oxygen consumption, smaller total lung capacities, large epiglottis, and anterior cephalad larynx [11,12]. Because inadequate preparation of anesthesia equipment further increases the risk of complications, preparing for pediatric anesthesia is considerably more demanding [13,14]. Therefore, pediatric anesthesiologists and nurses have to concentrate on more tasks during preparation, which is more likely to affect and limit their abilities, including memorization [15]. These factors can increase the risk of human error during the entire process from order creation to communication and the application of pediatric anesthesia.

Although fail-safe measures such as checklists, electronic charts with real-time dosage assistance, and labeled prefilled syringes have been successfully adopted to reduce human error, the possibility of miscommunication remains problematic [16,17,18]. For example, severe drug overdoses can result from simple miscommunication [19].

Recent research has demonstrated several methods that use a structured checklist or communication tool to prevent human error due to miscommunication in medical practice. Nasiri et al. reported that implementing a structured handover checklist during intraoperative staff shift changes improved the quality of communication within the surgical team, lowered the information omission rate, and increased team satisfaction [20]. In addition, Randmaa et al. stated that implementing a communication tool in anesthetic clinics enhanced staff members’ perception of communication and safety climate and decreased the proportion of incident reports related to communication errors [21].

Considering this perspective, we assumed that implementing a mobile application (app) could enhance the quality of pediatric anesthesia by decreasing human error. In this study, we evaluated the efficacy of a mobile app in decreasing human errors by observing anesthesiologists and nurses performing pediatric anesthesia in the OR and collecting nurses’ responses.

## 2. Materials and Methods

This study comprised three parts: (1) the development of a mobile app, (2) an observational study to determine if the mobile app reduced human error, and (3) a survey of responses from nurses regarding the mobile app.

### 2.1. Mobile App Development

We developed an Android mobile app to generate and transfer orders to prepare for pediatric anesthesia. While developing the application, we had the following three goals: (1) the app should be easy to use and straightforward to generate orders quickly, especially during emergencies, (2) we aimed to eliminate miscalculations of drug doses; therefore, we designed it such that users do not need to calculate anything, and (3) we worked hard to minimize miscommunication between doctors and nurses so that all necessary orders were automatically generated and given to nurses in an easy-to-see format.

The app automatically generates the recommended medication regimen, endotracheal tube type, laryngeal mask airway, and angiocatheter size. Users can input body weight according to the patient’s age and select the type and dose of drugs, type of airway-securing device, arterial and venous line placement, and fluid. The app generates orders for nurses with regimens (e.g., mix 50 mg dopamine in 50 cc normal saline) and anesthesiologists with rate and dose (e.g., dopamine 0.1 mcg/kg/minute = 1.8 mL/h). Users can send the generated orders to other anesthesiologists and nurses by sharing the features through their mobile phones (Figure 1).

We began developing the app in November 2019 using Android Studio. We chose Kotlin as the programming language because it has simpler grammar and higher productivity than Java. In June 2020, we created a working version. The app is available on the Google Play Store (https://play.google.com/store/apps/details?id=com.lhm.pediatricanesthesia, accessed on 23 October 2023). We have used it in daily clinical practice and continuously updated it. We received ongoing feedback during development and use, and one of the biggest changes we made based on feedback was a differing order summary for nurses and doctors. Also, we added body weight of the patient in the summary for nurses by request from nurses.

### 2.2. Observational Study

#### 2.2.1. Study Design and Objectives

The primary objective of this study was to determine whether the mobile app can reduce human error in pediatric anesthesia settings. After developing a working version of the app, we recorded the frequencies and types of errors made by anesthesiologists or anesthesia nurses related to the creation, preparation, and execution of pediatric anesthesia orders in the electronic medical records (EMR). We recorded the data for pediatric general anesthesia only. We excluded non-general anesthesia and short surgeries (less than 30 min) because they were central line changes or a chemoport removal in which the preparation was routine and the use of the application was not feasible. We did not observe human errors during emergency anesthesia because multiple nurses’ involvement increased the risk of incorrect observation. An anesthesiologist was considered to have made an error when they modified an order already given to the nurse or made an additional order that they had forgotten to make. If medication or equipment was not ready because the nurse did not correctly follow the order received, then each piece of medication or equipment was regarded as an error. We encouraged the use of the mobile app, but the final decision was up to the anesthesiologist in charge. We retrieved EMR data from July 2020 to January 2022 after obtaining permission from the Data Review Committee of the hospital (Approval Number: 20220203-F-021) and the institutional review board (IRB) (KC22EISI0198, date of approval: 6 May 2022).

We collected data on the age of the patient, duration of anesthesia, type of anesthesiologist (resident, non-pediatric anesthesiologist consultant, pediatric anesthesiologist consultant), years of experience as an anesthesia nurse, the use of the mobile application, number of additional orders from the anesthesiologist, number of modifying orders after inducing anesthesia, incidence duration during which a nurse left the operating room before induction of anesthesia due to a change in the anesthesiologist’s order or nurse error, incidence and period during which a nurse left the operating room after induction of anesthesia due to a change in the anesthesiologist’s order or nurse error, number of times nurses missed orders, number of dosing errors, incidence, and total amount of delay in surgery due to order-related human error.

#### 2.2.2. Sample Size Calculation

We could not find prior studies on human error reduction in pediatric anesthesia preparation. Therefore, while developing the mobile application, we investigated the rate at which anesthesiologists make corrections/changes to the order while creating and delivering the order to the nurse. The preliminary result was 29% with the traditional method and 12% using the app we were developing. We calculated the sample size using a 2-sided Fisher’s exact test at a significance level of *p* < 0.05 with 80% power. The required sample size was 96 patients per group. We set the experimental group as 101 participants with a 5% margin.

#### 2.2.3. Bias and Confounding Factors

We assumed that the patient’s age, the duration of the anesthesia, and the type of anesthesiologist were potential confounding factors, and the nurse’s number of working years was an effect of modifying factors. We matched the App group to the Conventional group by dividing the patient’s age into less than 1 year and equal or older than 1 year (neonate and infant vs. older child), the duration of anesthesia into less than 2 h and equal or longer than 2 h (simple vs. complex surgery), and the nurse’s number of working years into less than 3 years, 3–5 years, 5–10 years, and 10 years or more (level of experiences).

#### 2.2.4. Statistical Analysis

The IBM SPSS Statistics for Windows (Version 28.0. Armonk, NY, USA, IBM Corp) was used for statistical analysis. All the tests were two-sided. Differences were considered statistically significant if the *p*-value was <0.05, which are marked with an asterisk (*).

We used the mean and the difference between the means and the 95% confidence interval (CI) or the median (interquartile range) to present the results. The difference between the median and the 95% CI of the difference between the medians was based on the Hodges–Lehmann method. Differences between groups were analyzed using Student’s *t*-test, Mann–Whitney U test, or Fisher’s exact test, when appropriate. Odds ratios (ORs) with 95% CIs were used to represent Fisher’s exact test.

We assumed that the age of the participant, the duration of the anesthesia, and the nurse’s working years were potential confounding factors. We used logistic univariate analysis to examine the association between potential confounding factors and the incidence of human error among anesthesiologists and nurses. We then performed a multiple logistic regression analysis using the stepwise selection method with these covariates to reduce the confounding and bias. We calculated adjusted odds ratios (aORs), 95% CIs, and *p* values.

### 2.3. Survey

A survey was conducted that investigated the nurses’ perceptions of pediatric anesthesia patient safety and ease of preparation when receiving orders via the anesthesiologist-developed mobile app. Because only a few anesthesiologists participated in the pediatric anesthesia, a survey for the anesthesiologists was not feasible. The IRB approved the study, and the authors followed the principles of Good Clinical Practice. The questionnaire comprised 15 questions that took 5–10 min to complete. After obtaining approval from the IRB (KC22EISI0198, date of approval: 6 May 2022), all anesthesia nurses at a single tertiary hospital received the questionnaire in July 2022. We requested a total of 63 anesthesia nurses to participate in the survey, of which we received 50 valid responses. Descriptive statistics (absolute frequencies and percentages) were analyzed using IBM SPSS Statistics for Windows (version 28.0. Armonk, NY, USA, IBM Corp). The participants provided informed consent at the beginning of the survey. Survey participation was voluntary. The questionnaires were collected unattended, and the participants placed the completed questionnaires in a secure box so that their participation was anonymous. The entire questionnaire can be accessed in Appendix A.

### 2.4. Ethics

This study was conducted in accordance with the guidelines of the Declaration of Helsinki. The study protocol was reviewed and approved by the IRB of St. Mary’s Hospital, Seoul, Korea (KC22EISI0198, date of approval: 6 May 2022) and registered at the Clinical Research Information Service (http://cris.nih.go.kr, accessed on 23 October 2023; KCT0007357). Retrospective data were collected after obtaining approval from the hospital’s Data Review Committee (Approval Number: 20220203-F-021). Informed consent was obtained from the parents or legal guardians of all patients.

## 3. Results

### 3.1. Observational Study

#### 3.1.1. Study Design

The current study is an observational retrospective matched cohort single-center study. During the study periods, we retrieved 813 cases of pediatric anesthesia. After applying the exclusion criteria (non-general anesthesia, short surgery time, incomplete data, or emergent surgery), 507 cases were still eligible. After matching for the duration of anesthesia, nurse working years, and patient age, there were 101 cases in the App group and 101 in the Conventional group (Figure 2).

Baseline Characteristics before and after matching are presented in Table 1.

#### 3.1.2. Anesthesiologist Errors

Anesthesiologists were less likely to add or modify an existing order in the App group. The unadjusted OR was 0.26 (95% CI:0.141–0.533, *p* < 0.0001 (*)). The adjusted odds ratio was 0.158 (95% CI = 0.074–0.337, *p* < 0.01 (*)). The likelihood of an anesthesiologist adding or changing an order while preparing for pediatric anesthesia increased when the patient’s age was less than one year or when preparing for anesthesia longer than 2 h (Table 2).

The difference between means was −0.44 (0.64 vs. 0.20, 95% CI: −0.603 to −0.231, *p* < 0.0001 (*)). The incidence of three or more errors during the anesthesia order generated by anesthesiologists was 2.9% in the Conventional group and 0% in the App group, as shown in Table 3.

#### 3.1.3. Errors Communicating Orders between Anesthesiologists and Nurses

The incidence of making errors from anesthesia nurses’ taking orders from anesthesiologists was lower in the App group. The unadjusted OR was 0.47 (95% CI: 0.211–0.948, *p* = 0.0323 (*)). The adjusted OR was 0.458 (95% CI: 0.220–0.955, *p* = 0.037 (*)).

The total number of errors during the order transfer between the anesthesiologists and nurses was also lower in the App group (0.25 in 0.14, *p* = 0.0435 (*)). The difference in the means was 0.11 with a 95% CI of 0.0034–0.2292. There was one instance in which two errors occurred in the Conventional group while communicating the order but none in the App group, as shown in Table 3.

#### 3.1.4. The Frequency of Anesthesia Nurses Leaving the OR

The frequency of order-related leaves from the OR of anesthesia nurses was lower in the App group. The unadjusted OR was 0.39 (95% CI:0.111–0.676, *p* = 0.0021 (*)). The adjusted OR was 0.237 (95% CI:0.120–0.467, *p* < 0.001 (*)). In the Conventional group, the anesthesia nurse left the OR three or more times in 2% of cases, compared to 0% in the App group, as shown in Table 3.

#### 3.1.5. Total Duration of Anesthesia Nurses Leaving the OR

The total time of leaving the OR was shorter in the App group (3.55 vs. 1.47 min, *p* = 0.0011 (*)). The difference between means was −2.08 with 95% CI −3.02 to −0.98. In the Conventional group, there were 2% of cases where the total time out of the OR was 15 min or more, but 0% in the App group, as shown in Table 4.

#### 3.1.6. Delays of Surgery Due to Anesthesia Order-Related Problems

The incidences of delays in surgery due to anesthesia order creation, delivery, and preparation were 18.4% and 9.9% in the Conventional and App groups, respectively. However, this difference was not statistically significant (OR 0.47, 95% CI, 0.207–1.011; *p* = 0.1247). The delay time was also not significantly different between the two groups (1.49 vs. 0.78, *p* = 0.0731).

### 3.2. Survey

Of the 63 questionnaires distributed, 52 were returned; two were excluded from the analysis due to incomplete responses (Figure 3).

The surveyed anesthesia nurses that had worked for <3 years was 30%, 3–5 years was 12%, 5–10 years was 20%, or ≥10 years was 34%. When asked about the necessity for correction of conventional verbal orders, the anesthesia nurses responded as follows: no change, 16%; one or two corrections per anesthesia, 76%; three or more corrections per anesthesia, 4%; and constant changes, 4%. However, the collected data showed that the rate of one or two corrections per order was 31.2%, which was much lower than the rate reported in the survey responses (76%).

The next question was “How many times during the preparation of pediatric anesthesia do you have to double-check with the anesthesiologist due to unclear, incorrect, or missing orders?”. The responses were 40% for at least every round of anesthesia, 30% for every two rounds of anesthesia, 24% for every three or four rounds of anesthesia, 6% for every five or more rounds of anesthesia, and 0% for never. The 40% rate of at least one double-check per anesthesia reported by nurses was similar to the 45% rate of order modification/addition per anesthesia seen in the data we collected.

Responses to the time needed to prepare for pediatric anesthesia varied as follows: <5 min was 6%, 5–10 min was 20%, 10–15 min was 34%, 15–20 min was 26%, and >20 min was 14%.

When asked how often they left the operating room during a round of anesthesia because of a change or new order from the anesthesiologist, 14% said zero times, 44% said once, 36% said twice, and 6% said three or more times. The number of leaves felt by the nurses is much higher than the observed data of 47.1% in the Conventional group and only 24.8% in the App group who left the OR at least once during anesthesia in the data collected.

When asked about the duration of time nurses left the room during each round of anesthesia due to changes or additions to the anesthesiologist’s order, 56% of the nurses chose less than 5 min, 30% chose 5–10 min, 8% chose 10–15 min, 2% chose 15–20 min, and 4% chose more than 20 min. Anesthesia nurses thought that they left the OR longer than the observed amount of time, which was 80% in the Conventional group and 96% in the App group, within 5 min.

Seventy-six percent of the nurses surveyed said that they had received orders through the app. Looking at responses from nurses who used the app order, 87% said it was easier to prepare with the app and 95% said it was more effective in reducing medication errors.

Of those who received orders using the app, 68% reported no order modifications and 32% reported one to two changes in orders. In our data, an order revision was unnecessary in 82% of anesthesia cases, and 18% needed one to two revisions. The number of nurses who reported experiencing three or more order revisions was zero. This was also true of the data collected.

When anesthesia nurses were asked how often they had to ask the anesthesiologist to clarify unclear, incorrect, or missing orders after receiving an app order, 3% said they had to query once for each round of anesthesia, 27% said once per two anesthesia, 15% said once per three to four anesthesia, 27% said once per five or more anesthesia, and 27% said that no reconfirmation was needed.

Approximately 95% of anesthesia nurses with app order experience thought that using the app could decrease the frequency of leaving the OR during anesthesia. Additionally, approximately 90% responded that the app helped to reduce their duration of leaving the OR.

When asked about the convenience of preparation of anesthesia among different order methods on a scale of 1–5 (where 1 = “very inconvenient” and 5 = “very convenient”), nurses rated the app as convenient or very convenient at 86%, followed by EMR at 20% and verbal orders at 6%. The percentages of respondents who were inconvenienced or very inconvenienced were EMR (60%), verbal (44%), and app (6%), respectively, as shown in Table 5.

When anesthesia nurses were asked about the likelihood of errors based on the order delivery method on a scale of 1–5 (where 1 = more errors and 5 = fewer errors), fewer and fewer errors were chosen in the app order (88%), EMR order (38%), and verbal order (12%). Nurses thought that many or more errors were associated with verbal orders (44%), EMR orders (34%), and app orders (2%) as shown in Table 6.

## 4. Discussion

This study showed that mobile application-based order creation and communication decreased human error by both anesthesiologists and nurses during pediatric anesthesia preparation. The risk of an anesthesiologist committing an error while creating orders for the preparation of pediatric anesthesia was reduced by more than 70% when using the mobile app compared with a conventional verbal order. The results of this study align with previous studies showing that assistive technology can help doctors and nurses prevent human error and increase patient safety. A recent Cochrane review showed that computerized assistance may reduce medication errors by 25%, and this effect was more profound with an advanced computerized system [22]. The effect of our app’s error prevention was much stronger than the reported effects of computer assistance. It shows that the traditional method was threatening patient safety, and on the other hand, the app is highly effective in preventing human error. The strong error-preventing effect of our mobile app is likely because it was designed and developed entirely by a pediatric anesthesiologist. Order assist or error preventive computerized system is usually developed by non-medical personnel and it is not just for anesthesia but for an entire hospital’s medical system. It may have more powerful features, but it may not be as effective at preventing human error in anesthesia preparation as one developed by anesthesiologists. From the very beginning of the app’s development, our priority was to avoid human errors while using the app. To perform this, we ensured that the order creation process did not require any calculations at all and that as much of the content as possible was automatically generated or could be created with a single click. After entering bodyweight and age, all you have to do is select your medications, equipment, and intravenous fluid, just like you would choose food at a kiosk at a fast food restaurant. This makes the app very intuitive to use and requires very little learning. Although we tried to make the app as easy as possible, the app only contains information about medication and equipment available to the author’s institution. However, it still could be useful in other clinical settings because we included as many basic medications and equipment as possible. Also, the app’s recommendation of the proper depth of the endotracheal tube, type of intravenous lines and depth of insertion could be useful outside the author’s institution.

Communication errors between anesthesiologists and nurses decreased by approximately 50%. Communication is one of the principal “non-technical skills” which is “the cognitive, social, and personal resource skills that complement technical skills, and contribute to safe and efficient task performance” [23]. Training can improve communication skills for encoding, transmitting, and receiving information. For example, the practice of read-back improves the accuracy of communication [24]. However, when preparing for pediatric anesthesia, both anesthesiologists and nurses exchange a tremendous amount of information in a relatively short period and read-back to every item is practically impossible. Another challenge is training new nurses. Multiple training sessions are required to become proficient in non-technical skills, which requires time [25]. According to the survey results, the distribution of anesthesia nurses in our hospital according to their length of service was U-shaped. Nearly 40% of nurses have less than five years of work experience, and the risk of human error could be higher.

Our app could offer a solution to this potential issue because it helps send and receive orders in an organized manner without training. When anesthesiologists finish the selection of orders, the app shows a list of orders for nurses, which presents a dose and mixing regimen of medications along with the types of equipment required. Anesthesiologists can double-check the final list and send it to a nurse. Nurses can carry a mobile phone to the anesthesia preparation room and collect the required equipment and medicine, as presented on the phone. Thus, the app reduces the need for talking, memorizing, or writing while communicating orders for pediatric anesthesia preparation.

The frequency of anesthesia nurses leaving the OR to visit anesthesia preparation rooms due to order changes, new orders, or miscommunication was reduced by 60%. Also, the mean time spent leaving the OR was shorter. Although the difference between the means was only two minutes, the percentage of nurses leaving the OR for more than 10 min was 6.86% in the Conventional group compared to 0.99% in the App group. It appears that nurses were able to identify the items they needed more efficiently in the anesthesia preparation room when using the app. However, the incidence of delayed surgery due to anesthesia preparation was not statistically significant.

According to the survey responses, anesthesia nurses perceived anesthesiologists’ order changes to be more frequent than they actually were, and both the perceived number of times and length of time they were out of the OR owing to a changed order were greater than they actually were. This demonstrates the psychological burden on anesthesia nurses regarding orders that are added or changed during anesthesia.

More than 90% of the nurses who experienced app-based orders agreed that it could reduce the incidence and duration of nurses’ leave from the OR during anesthesia, which is in line with the data collected.

Of the nurses with experience with the app, 86% thought it was convenient for preparing pediatric anesthesia, and 88% believed that the use of the app was associated with fewer errors.

This study has several limitations. First, this was an observational study and there could have been a selection bias. Initially, the authors intended to conduct an additional double-blind prospective clinical study but after seeing the results of this study, the authors felt it would be unethical to the patients. Currently, the authors are using the app for all pediatric anesthesia. To reduce bias, we matched the groups with duration of anesthesia, nurse working years, age of the patients, and types of anesthesiologists. However, matching does not eliminate confounding effects. Sometimes, it causes additional confounding problems that do not exist in the source population [26]. Second, counting errors, frequencies, and durations could be omitted or incorrectly recorded. The data collection for human error relied on the attending anesthesiologist. To reduce bias, the records were checked daily. Third, this study included only elective surgeries. Because preparing anesthesia for emergency surgeries usually involves multiple nurses and makes accurate data collection difficult. However, the authors have used the app in pediatric emergency surgery preparation without problems. Although this study was limited to the preparation of elective surgeries at one hospital, we developed the application to be as universally and easily applicable as possible. Therefore, we believe it will show similar effects when used in other hospitals, even in emergency situations. Fourth, we did not record the actual number of orders. Fifth, we divided each potential confounder into two or three groups for matching and analysis. We categorized age as before or after one year of age because infants and toddlers are physiologically/anatomically more fragile than older children [27]. For the duration of surgery, we categorized it with the two-hour cutoff based on a review that showed surgery longer than two hours is associated with increased complications [28]. For nurses’ working years, we categorized them based on 3, 5, and 10 years, which is a role change year (specialty anesthesia, night chief) for the anesthesia nurses in our hospital. These categorizations could be a source of bias. Sixth, because nurses knew the developer of the mobile app, it is possible that the survey results were skewed in favor of the app developer, who was a pediatric anesthesiologist at the same hospital. Finally, this study was conducted at a single tertiary hospital. The app used in this study was optimized for pediatric anesthesia at a specific institution. However, the developers have tried to make it as universally usable as possible

Currently, the app’s functions are limited to preparing for pediatric anesthesia, but we plan to develop features that anesthesiologists and nurses can refer to in cases of pediatric anesthesia-related emergencies (hypoglycemia, malignant hyperthermia, anaphylaxis, etc.).

## 5. Conclusions

This study showed that the use of an order-support mobile app developed by pediatric anesthesiologists was associated with more than half of the reduction in the incidence of human error among anesthesiologists and nurses during preparation of pediatric anesthesia.

Nurses who have experience with the app-based order agree that it is more convenient and less error-prone than conventional methods.

## Figures and Tables

**Figure 1 children-10-01860-f001:**
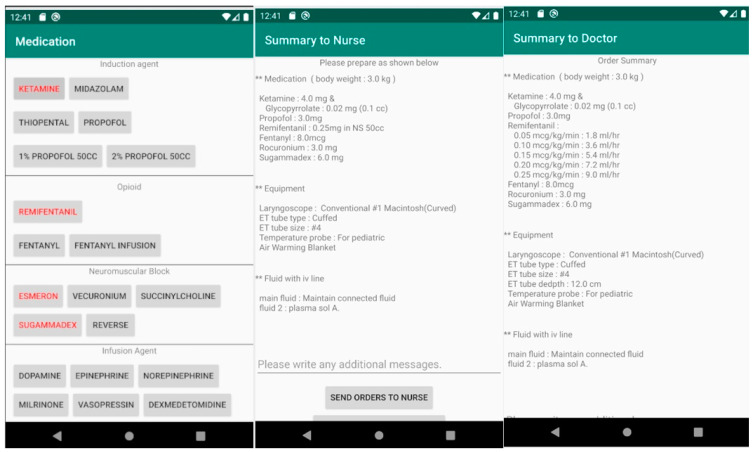
Screenshot of the mobile application. Red text on a button: The medication is selected.

**Figure 2 children-10-01860-f002:**
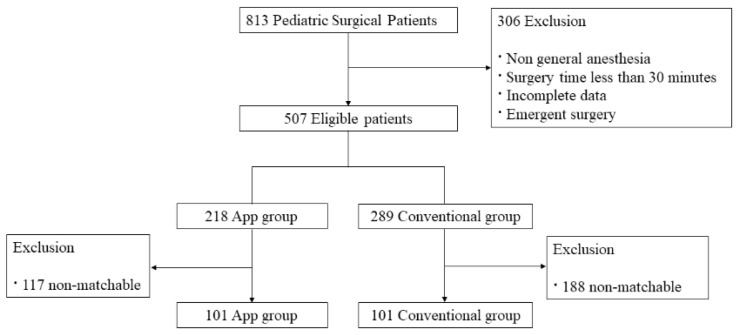
Eligibility screening flow diagram.

**Figure 3 children-10-01860-f003:**
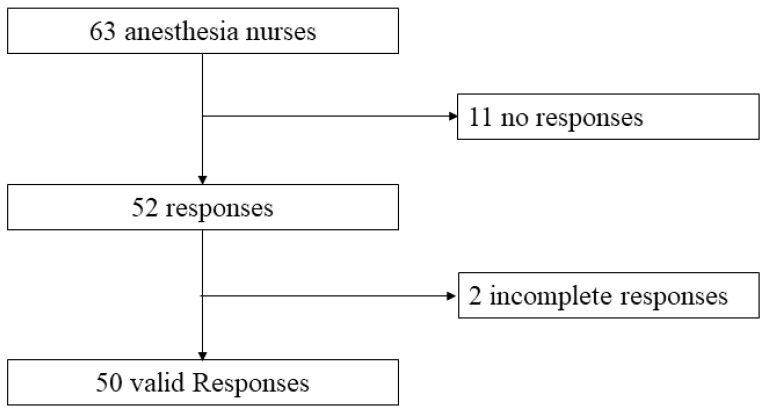
Survey response flow diagram.

**Table 1 children-10-01860-t001:** Baseline Characteristics.

	Before Matching (*n* = 507)		After Matching (*n* = 202)	
Variable	App Group (*n* = 218)	Conventional Group (*n* = 289)	App Group (*n* = 101)	Conventional Group (*n* = 101)
Age (days)				
<365 days	63.8 ± 2.9	92.6 ± 4.2	75.6 ± 2.3	78.9 ± 2.7
≥365 days	523.7 ± 1134.9	947.9 ± 1422.7	619.2 ± 989.2	633.9 ± 1141.3
Sex (M/F)				
	135/83	151/138	88/13	64/37
Duration of Surgery (minutes)				
<120 min	82.4 ± 16.6	77.4 ± 19.9	75.6 ± 15.1	78.9 ± 17.5
≥120 min	308.6 ± 178.5	213.6 ± 144.2	263.5 ± 137.5	230.0 ± 126.8
Nurse Working Years (years)				
	89/19/28/82	121/22/40/106	44/3/10/44	46/4/14/37
<3 years	2.5 ± 1.1	2.6 ± 0.9	2.5 ± 0.9	2.5 ± 0.8
3–5 years	3.8 ± 0.7	4.1 ± 1.0	3.9 ± 0.9	4.0 ± 0.9
5–10 years	7.4 ± 3.4	6.9 ± 4.1	7.2 ± 2.3	7.1 ± 3.5
>10 years	13.4 ± 4.4	12.6 ± 5.8	12.8 ± 3.4	12.7 ± 6.1

**Table 2 children-10-01860-t002:** Multiple logistic Regression Analysis for Anesthesiologist errors. (*): *p*-value < 0.05.

Variables	OR	95% CI	*p*-Value
Use of App	0.158	0.074–0.337	<0.001 (*)
Age (≥1 year)	0.411	0.193–0.877	0.021 (*)
Duration of Anesthesia (≥2 h)	2.943	1.445–5.994	0.003 (*)

**Table 3 children-10-01860-t003:** Frequency of Errors and Anesthesia nurse’s leave.

		App Group (Number (%))	Conventional Group (Number (%))
Anesthesiologist Errors			
Frequency	0	83 (82.2%)	56 (55.4%)
	1	15 (14.9%)	28 (27.7%)
	2	3 (2.9%)	14 (13.9%)
	≥3	0 (0%)	3 (3.0%)
Errors while communicating orders			
Frequency	0	87 (86.1%)	77 (76.2%)
	1	14 (13.9%)	23 (22.8%)
	2	0 (0%)	1 (1.0%)
Anesthesia nurse’s leaving the OR			
Frequency	0	76 (75.2%)	53 (52.5%)
	1	20 (19.8%)	36 (35.6%)
	2	5 (5.0%)	10 (9.9%)
	≥3	0 (0%)	2 (2.0%)

**Table 4 children-10-01860-t004:** Total duration of anesthesia nurse’s leave from the OR.

Time (Minutes)	App (%, Number)	Conventional (%, Number)
0	76 (75.3%)	53 (52.5%)
0–5	21 (20.8%)	27 (26.7%)
5–10	3 (2.9%)	14 (13.9%)
10–15	1 (1.0%)	5 (5.0%)
15–20	0 (0%)	1 (1.0%)
≥25	0 (0%)	1 (1.0%)

**Table 5 children-10-01860-t005:** Convenience of anesthesia preparation.

	App	EMR	Conventional
1	0%	20%	10%
2	6%	40%	34%
3	8%	20%	50%
4	30%	12%	4%
5	56%	8%	2%

Note: 1 = very inconvenient, 2 = inconvenient, 3 = average, 4 = convenient, 5 = very convenient. EMR: electronic medical records.

**Table 6 children-10-01860-t006:** Risk of human errors.

	App	EMR	Conventional
1	0%	16%	12%
2	2%	18%	32%
3	10%	28%	44%
4	26%	26%	10%
5	62%	12%	2%

Note: 1 = more errors; 2 = many errors; 3 = moderate; 4 = few errors; 5 = fewer errors. EMR: electronic medical records.

## Data Availability

The data presented in this study are available on request from the corresponding author. The data are not publicly available due to the hospital policy.

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
