# Peer review of "The Effects of an Order-Assist Mobile Application on Pediatric Anesthesia Safety: An Observational Study"

_children, 2023, doi:10.3390/children10121860_

Round 1
Reviewer 1 Report
Comments and Suggestions for Authors
The authors present an interesting study. However, we have some questions about the study, which must be answered before it can be accepted for publication.
- Provide eligibility criteria and sources and methods of participant selection.
- Clearly define all variables: response, exposures, predictors, confounders and effect modifiers.
- Specify all measures taken to address potential sources of bias.
-How was the sample size determined?
- Explain how the quantitative and qualitative variables were treated in the analysis.
- Describe the number of participants in each phase of the study.
- Describe the reasons for the loss of participants in each phase.
- Consider using a flowchart.
Results
Cross-sectional studies: describe the number of outcome events or provide summary measures
Main results - Provide unadjusted estimates and, where appropriate, adjusted for confounding factors, as well as their precision.
(e.g. 95% confidence intervals).
- Specify the confounders you are adjusting for and the reasons for including them.
- If you categorize continuous variables, describe the limits of the intervals.
- If pertinent, consider accompanying relative risk estimates with absolute risk estimates for a
relevant time period.
Discussion
- Summarize the main results of the study objectives.
- Discuss the limitations of the study, taking into account possible sources of bias or imprecision. Reason about both the direction and magnitude of any possible bias.
- Discuss the possibility of generalizing the results (external validity).
Reviewer 2 Report
Comments and Suggestions for Authors
I thank the editor for the opportunity to review this article, which offers very interesting perspective of using digitalization in pediatric anesthesia settings in order to increase patients’ safety. Please find bellow the main recommendations:
11. Lines 51-53. Please justify by wich means the pediatric anesthesia is more demanding
22. Introduction: the main error concerning pediatric anesthesia, as revealed by the introduction, seems to be the dificulty of the correct dosage. Please try to structure and present also other errors that, due a distubed communication flow, could appear. Which other factors (besides communication) could predispose to errors in pediatric anesthesia practice.
33. Methodology: Please provide a flowchart with the responders included in the survey section. It is not very clear the exact number of your responders and their role (anesthesiologists, nurses etc) (Abstract vs. Methodology sections lines 23 vs.137)
44. Did the survey address nurses only? Please provide a justification of excluding the anesthesiologist.
55. Please mention the randomisation technique used in the study.
66. Please mention if you included in the study only elective surgeries since only for this kind of procedures an exact planning is possible. How would you apreciate the use of the app in emergency situations since time concerns are even more relevant.
77. Statistical analysis. Please, clearly state the variables you compared and the statistical tests, respectively. A more detailed explanaition of wich tests for wich data were used.
88 Results. Please provide one table with the informations from the Tables 1-3, since the data presented here are qualitative data.
Comments on the Quality of English LanguageMinor editing of English language required
Reviewer 3 Report
Comments and Suggestions for Authors
The study presents a compelling case for the use of a mobile app in reducing errors during pediatric anesthesia. The introduction provides an adequate background and the references support the research effectively. The research design seems appropriate for the objectives, and the results are presented clearly. However, the methods section could be improved by providing more detailed information on how the data was collected and analyzed, which could enhance reproducibility.
While the conclusions are supported by the results, it would be beneficial to discuss potential confounders and limitations in more depth. For example, a discussion on how the app might perform in different hospital settings or under different usage conditions would be insightful.
Additionally, the manuscript would benefit from a more detailed explanation of the app's features and how it compares to other existing systems. It would also be prudent to include how user feedback from nurses was incorporated into the app's iterative development.
Lastly, there is a minor inconsistency in the Discussion where it mentions "hypoglycemia, malignant hyperglycemia," which seems like a typographical error and should probably read "malignant hyperthermia."
Round 2
Reviewer 1 Report
Comments and Suggestions for Authors
The authors made the recommended changes and the manuscript is now better. We suggest that the article can be accepted in the present version.
Reviewer 2 Report
Comments and Suggestions for Authors
Thank you for the revised form of the manuscript